# Usefulness of Cardiac Computed Tomography in Coronary Risk Prediction: A Five-Year Follow-Up of the SPICA Study (Secure Prevention with Imaging of the Coronary Arteries)

**DOI:** 10.3390/jcm11030533

**Published:** 2022-01-21

**Authors:** David Viladés-Medel, Irene R. Dégano, Isaac Subirana, Martin Descalzo, Mireia Padilla, Xavier Mundet, Francesc Carreras Costa, Xavier Alomar Serrallach, Anna Camps, Roberto Elosua, Jaume Marrugat, Rubén Leta Petracca

**Affiliations:** 1Cardiac Imaging Unit, Cardiology Department, Hospital de la Santa Creu i Sant Pau, Universitat Autónoma de Barcelona (UAB), 08193 Barcelona, Spain; mdescalzo@santpau.cat (M.D.); mpadillal@santpau.cat (M.P.); FCarreras@santpau.cat (F.C.C.); rleta@santpau.cat (R.L.P.); 2CIBER de Enfermedades Cardio-Vasculares (CIBERCV), Instituto de Salud Carlos III (ISCIII), 28029 Madrid, Spain; irene.roman@umedicina.cat (I.R.D.); isubirana@imim.es (I.S.); acamps@imim.es (A.C.); relosua@imim.es (R.E.); jaume@imim.es (J.M.); 3REGICOR Study Group, IMIM (Hospital del Mar Medical Research Institute), 08003 Barcelona, Spain; 4Faculty of Medicine, University of Vic-Central University of Catalonia (UVic-UCC), 08500 Vic, Spain; 5University Public Health Center El Carmel, Department of Medicine, Universitat Autonoma de Barcelona (UAB), 08193 Barcelona, Spain; 16274xmt@comb.cat; 6Barcelona Ciutat Research Support Unit-IDIAP Jordi Gol, redIAPP, 08007 Barcelona, Spain; 7Radiology Department, Clínica Creu Blanca, 08034 Barcelona, Spain; xalomar@creublanca.es

**Keywords:** coronary artery disease risk, cardiac computed tomography, coronary artery calcium, primary prevention

## Abstract

Accurate identification of individuals at high coronary risk would reduce acute coronary syndrome incidence and morbi-mortality. We analyzed the effect on coronary risk prediction of adding coronary artery calcification (CAC) and Segment Involvement Score (SIS) to cardiovascular risk factors. This was a prospective cohort study of asymptomatic patients recruited between 2013–2017. All participants underwent a coronary computed tomography angiography to determine CAC and SIS. The cohort was followed-up for a composite endpoint of myocardial infarction, coronary angiography and/or revascularization (median = five years). Discrimination and reclassification of the REGICOR function with CAC/SIS were examined with the Sommer’s D index and with the Net reclassification index (NRI). Nine of the 251 individuals included had an event. Of the included participants, 94 had a CAC = 0 and 85 a SIS = 0, none of them had an event. The addition of SIS or of SIS and CAC to the REGICOR risk function significantly increased the discrimination capacity from 0.74 to 0.89. Reclassification improved significantly when SIS or both scores were included. CAC and SIS were associated with five-year coronary event incidence, independently of cardiovascular risk factors. Discrimination and reclassification of the REGICOR risk function were significantly improved by both indexes, but SIS overrode the effect of CAC.

## 1. Introduction

In the next decades, an increasing incidence of acute coronary syndrome (ACS) is expected in Spain, mainly due to the aging of the population. Up to 60% of all ACS cases will occur in the elderly population by 2049 [1]. The most cost-effective way to control the significant health and mortality burden of cardiovascular disease (CVD) is to reduce its incidence by identifying high-risk populations. To improve the sensitivity and specificity of identifying high-risk individuals, imaging biomarkers could be included in CVD or coronary artery disease (CAD) risk functions [2,3].

Cardiovascular risk classification of asymptomatic individuals is performed using multifactorial functions based on sex, age, and traditional CVD risk factors. In Spain, the SCORE [4,5] and the REGICOR risk function are the most widely used. The REGICOR function is a calibration of the Framingham coronary risk function and was prospectively validated in the Spanish population [6,7]. Most CAD risk functions acceptably classify high-risk subjects. In contrast, moderate risk individuals (e.g., SCORE risk < 7.5% or REGICOR risk between 5–10%) include approximately 30% of ten-year cases of CAD in population aged 35 to 74 years [7,8].

Several studies have attempted to reclassify patients with moderate risk to a higher or lower risk category, by including other factors in the risk functions. These factors have included circulating, genetic, and subclinical biomarkers among others. The presence of carotid plaques [8], the concentration of C-reactive protein [9], and the ankle-brachial index [8,9,10,11] have an acceptable cost and have been shown to be independent predictors in CVD incidence. However, the addition of these variables to the CVD/CAD risk functions resulted in small improvements of function accuracy and reclassification, and results were not replicated in all studies.

The coronary artery calcium (CAC) score is the CAD non-invasive test with the highest discriminatory and reclassifying power, better than the other non-invasive tests [8,12] with limited evidence in asymptomatic patients. In primary prevention of diabetic patients, the presence, extension, and severity of coronary atherosclerosis by cardiac computerized tomography angiography (CCTA) improved CAD prediction models based on Framingham function [13]. However, in other studies of unselected asymptomatic patients, the addition of this atherosclerotic burden information to the Framingham function produced no clinically significant improvement [14].

The objective of the present study was to determine the association between CCTA derived scores and coronary risk. In addition, we wanted to determine if CCTA-derived scores improved coronary risk prediction in the Spanish population.

## 2. Methods

This was a prospective cohort study of 325 patients with no symptoms of CVD, recruited between 2013–2017 from 2 settings. A total of 231 underwent a CCTA exam, as part of a general health check in a private clinic, while 94 were included in a primary care center (Figure 1).

The inclusion criteria were: age between 55–74 years, absence of previous CV symptoms, or overt disease. The exclusion criteria were: allergy to iodinated contrast, advanced renal failure with glomerular filtration rate <30 mL/min/1.73 m^2^, body mass index >40 kg/m^2^, inability to perform an adequate breath-hold apnea, or to have a non-conclusive CCTA due to the presence of artifacts.

### 2.1. Cardiac Computed Tomography Acquisition Protocol

CCTA exams were performed with two different scanners: Aquilion One with 320-row (Canon Medical Systems, Otawara, Japan) and iCT of 256-slice (Philips Healthcare, Amsterdam, The Netherlands). The study protocol began by performing a topogram to determine the limits of cardiac volume acquisition, generally limited between the tracheal carina, trachea, and diaphragmatic domes. The irradiation parameters were adjusted to the morphological characteristics of each patient: tube voltage and current of 100–120 kV and 250–400 mA, respectively. The CCTA studies were performed after administration of iodinated contrast by an infusion pump through an antecubital vein using a contrast volume of 1 mL/kg of iobitridol (Xenetix 350 mg/mL, Guerbet, Aulnay-sous-Bois, France) followed by the infusion of 40 mL of saline, both infused between 5–6 mL/s. The study was acquired using a bolus-track technique with a region of interest located in the ascending aorta. The CAC study was performed according to the internationally agreed protocol [15] and the epicardial arterial calcium burden was assessed with the Agatston score (AS).

Images were analyzed by two level 3 cardiac computed tomography readers who interpreted the data. Coronary arteries were assessed using a 16-segment model (modified 15-segment model with segment 16 being the intermediate branch). Luminal diameter stenosis severity was graded as 0% (none), 1–29% (very mild), 30–49% (mild), 50–69% (moderate), 70–99% (obstructive), and 100% (totally occluded). The segment involvement score (SIS) was calculated to determine the extent of coronary atherosclerosis as a measure of coronary atherosclerotic burden. The SIS score, ranging from 0 to 16, was calculated as the total number of individual coronary artery segments exhibiting plaques irrespective of the degree of luminal stenosis or its composition. The effective radiation dose of the CCTA and the CAC study was calculated using the dose-length product and the conversion factor for thorax (k = 0.014 mSv/ (mGy × cm) [16].

### 2.2. Follow-Up and Outcome of the Study

Participants were followed-up for a median of 4.6 years (1.1–5.9 years). The outcome included the following events: acute myocardial infarction (AMI), angina, coronary angiography, and/or revascularization. Events were obtained from the electronic medical records and by linkage with the hospital discharges database (CMBD) and the mortality register through the Catalan Program on Analytical Data for Research and Innovation in Health (PADRIS). Participants without follow-up data were excluded.

### 2.3. Measurements

Demographic characteristics (age and sex) and CV risk factors (total and HDL cholesterol, systolic and diastolic blood pressure, diabetes, and smoking) were obtained from the electronic medical records. CAD risk was calculated with the 10-year REGICOR risk function [6,17].

### 2.4. Statistical Analysis

Participant characteristics at baseline were described by event incidence and by categories of the CAC and the SIS. CAC and SIS were included as continuous and categorical variables in the descriptive analysis. CAC categories were: AS = 0 Hounsfield units (HU), AS = 1–99 HU, AS = 100–300 HU, and AS > 300 HU. SIS categories were: SIS = 0 segments with plaque, SIS = 1–4, and SIS > 4. CAC and SIS were also dichotomized as CAC = 0/CAC > 0 and as SIS = 0–4/SIS > 4. Participant characteristics were summarized by the mean, standard deviation and a t-test for continuous variables, and by frequencies and chi-squared or Fisher exact test for categorical variables. The standardized differences were also calculated. Crude association between variables and event incidence in the follow-up was examined with time-dependent log-rank test. Pearson correlation coefficient was computed to assess the association between CAC/SIS and Framingham-REGICOR risk.

The adjusted hazard ratio (HR) of event incidence for CAC/SIS categories was analyzed using Cox proportional hazard models adjusted for the ten-year REGICOR risk. Four models were built with the following predictor variables: REGICOR risk alone; CAC and REGICOR risk; SIS and REGICOR risk; and CAC, SIS, and REGICOR risk. CAC and SIS were included as continuous variables in the Cox models because the association between CAC/SIS and the outcome was linear and the non-linear component was not significant in models with splines. We also did an analysis with SIS as a categorical variable with a cutoff at SIS = 5. The CAC HR was estimated for a 100 arbitrary unit increase. Discrimination was examined with Sommer’s D index to consider censoring. Reclassification was assessed with the continuous and categorical net reclassification index (NRI) [18] considering censoring. For the categorical NRI, cutoff points were based on the REGICOR risk function. The REGICOR function defines low, moderate, and high-risk cutoff points at 5 and 10% in 10-year risk. The categorical NRI cutoff points were set at 2.5 and 5% to approximate the prediction to the available follow-up, whose maximum was close to 5 years. NRI confidence intervals were obtained by bootstrapping. NRI was computed for the whole cohort and for cases and non-cases. The categorical NRI was also calculated for participants at intermediate Framingham-REGICOR risk (5–10%), which is commonly known as clinical NRI.

Analyses were performed with R statistical software version 3.5.2 (R Core Team, Vienna, Austria) [19].

### 2.5. Ethical Aspects

Participants signed an informed consent at the time of inclusion that contained, among others, authorization for subsequent clinical follow-up. The ethics committee of the Hospital de la Santa Creu i Sant Pau approved the study protocol and the CAC and CCTA results were sent to the medical staff responsible for the primary CVD prevention of the participants. The decision to modify the medical treatment and/or perform further explorations was made at the discretion of the treating physician.

National and international human study ethical guidelines (Seventh Revision of the Helsinki Declaration—World Medical Association (2013) [20] and the Spanish legal regulations on the confidentiality of personal data) were followed.

## 3. Results

Baseline characteristics of the included participants are presented in Table 1 and distribution of participants by REGICOR risk and by CAC/SIS categories is shown in Appendix A.

Nine of the 251 individuals included in the study had a coronary event in the follow-up (2 AMI with ST segment elevation, 3 AMI without ST segment elevation, 2 coronary angiographies, and 2 revascularizations) (Figure 1). Participants developing CAD had lower high-density lipoprotein cholesterol and higher systolic blood pressure, REGICOR risk, CAC, SIS, and diabetes prevalence (Table 1). Of the 251 study participants, 94 had a CAC = 0 and 85 a SIS = 0. Participants with higher CAC were older, more frequently men, had lower total and HDL cholesterol, and higher systolic blood pressure, REGICOR risk, SIS, diabetes prevalence, and CAD incidence in the follow-up (Appendix A). Participants with higher SIS were older, more frequently men, had lower total and HDL cholesterol, and higher systolic and diastolic blood pressure, REGICOR risk, CAC, diabetes prevalence, and CAD incidence in the follow-up (Appendix A).

Mean REGICOR risk increased from 9.04 in participants with CAC = 0 to 17.7 in those with CAC > 300 HU, and from 8.45 in participants with SIS = 0 to 17.4 in those with SIS > 4 segments with plaque (Appendix A). Three participants with low REGICOR risk had CAC higher than 300 and 12 participants with high REGICOR risk had CAC = 0 (Appendix A). Pearson correlation coefficient of REGICOR risk and CAC was 0.28 [95% confidence interval (CI) 0.15, 0.40]. Six participants with low REGICOR risk had SIS > 4 and 10 participants with high REGICOR risk had SIS = 0 (Appendix A). Pearson correlation coefficient of REGICOR risk and SIS was 0.49 [95% CI 0.38, 0.59].

Participants with CAC = 0 or SIS = 0 had no CAD events in the follow-up (Appendix A). The percentage of participants with CAD events increased from the participants with lowest CAC and SIS to the ones with highest scores. In separate models and after adjusting for REGICOR CAD risk, the HR of CAD events was 1.07 [95% CI 1.03, 1.11] for 100 units increase of the CAC, and 1.42 [95% CI 1.18, 1.78] for each coronary segment with plaque of the SIS. In a third model combining both scores adjusting for the REGICOR CAD risk, only SIS remained significant to predict CAD events (1.40 [95% CI 1.10, 1.78]). In the model with SIS as a categorical variable, having a SIS > 4 was associated with increased incidence of CAD events, although it did not reach statistical significance (1.29 [95% CI 0.91, 1.84]; *p* = 0.158). Discrimination and reclassification analyses are presented in Table 2 and Table 3. Discrimination Sommer’s D index increased significantly from 74 to 89%, with the addition of SIS or of SIS and CAC to the REGICOR CAD risk (Table 2). Reclassification measured with the continuous NRI improved when the SIS, CAC, or both were added (Table 3). This improvement was observed for the whole cohort and for non-cases. The categorical NRI yielded an increase in reclassification by the inclusion of SIS and of both scores. The categorical NRI also improved in the whole cohort and in non-cases (Table 3). The reclassification of participants with intermediate REGICOR CAD risk was significantly improved by the addition of both scores, separated and together.

## 4. Discussion

In the present study, we showed that information from non-invasive coronary arteries imaging is associated with CV events, correlates with CAD risk, and its addition to CV risk factors improves CAD prediction and reclassification. Calcified coronary plaque phenotype using CAC score has been widely used in asymptomatic individuals and shown to provide an important CVD incidence prognostic information in various age groups, gender, CVD risk factors, and ethnicities [21,22,23]. There was, however, scarce evidence on whether SIS added prognostic information beyond CAC and traditional CVD risk factors.

The more important contributions of the present study are the distribution of CAC and SIS across different CAD risk levels in asymptomatic population, and the improvement in CAD risk prediction and reclassification when adding this information to a CAD risk function. This improvement was greater adding SIS than when adding CAC, which differs from previous studies [24].

The imperative to screen for CVD is well recognized because the progression of atherosclerosis occurs over decades and there is usually a long clinical latency before cardiac symptoms are observed. The reliance on global algorithms based on subjects’ risk factors is limited because they do not establish a diagnosis of CAD or CVD, but rather provide a probabilistic assessment of CVD likelihood according to group data. The REGICOR risk function is one of the most used risk functions to estimate ten-year CAD risk in primary prevention of asymptomatic Spanish population. Validation studies [7,8] showed that approximately 30% of CVD events occur in moderate-risk patients (5–10% at ten years in REGICOR and 10–20% in Framingham) who are usually not considered for intensive therapeutic management, which revealed one of the limitations of CVD risk functions in general. In the present study, we found that NRI (i.e., patients correctly reclassified to higher or lower risk group) among those with moderate CV risk adding CAC to the REGICOR function was significantly improved.

This study also confirms the good CVD prognosis in a mid-term follow-up of individuals with CAC and SIS = 0 as none of 94 and 85 subjects, respectively, experienced a CAD event in the follow-up. This result suggests a high specificity of a CAC and SIS 0 value. In addition, these scores may help to guide discussions regarding identification of individuals less likely to receive net benefit from lifelong preventive pharmacotherapy [25].

Unlike CAC, SIS also accounts for non-calcified plaque, which could be the substrate for MACE [26], but its prognostic impact over CAC plus CV risk factors remains uncertain [24,27]. In our study, SIS (a simple measure of the extent of coronary atherosclerosis) has independent prognostic value that seems to be comparable to CAC. For each coronary segment affected, we found a higher HR for MACE than previously reported (1.45; 95% CI 1.19, 1.78) [28,29], although events reported in previous studies (death or MI) differ from the events considered in the present study (MI, angina, coronary angiography, and/or revascularization). SIS shows better correlation with ten-year REGICOR CAD risk score, and higher reclassification power according to MACE than CAC. SIS offers a numeric quantification of the number of segments affected by atherosclerosis, which perhaps could be used in the future as a reference for comparison of CCTA investigations for a given asymptomatic patient and assist in the quantification of degree of plaque progression.

Determination of a SIS threshold that would predict MACE may be useful in the clinical setting and would be helpful for direct comparison with the binary evaluation of presence of obstructive CAD. Our study found a HR = 1.29 (95% CI: 0.91–1.84, *p*-value = 0.158) in those individuals with SIS > 4 and non-obstructive CAD, which is similar to previously reported [30] and similar for the presence of obstructive CAD and SIS < 4.

Patients with CAC and/or non-obstructive CAD still have substrate for MACE, regardless of their CV risk by REGICOR function. Patients with non-obstructive CAD demonstrated by CCTA, especially those with SIS > 4, may benefit for statin therapy in the follow-up [31,32]. We must also highlight that coronary CCTA has been associated with improved medical treatment compliance and tighter CAD risk factor control [33,34]. Even so, more studies are needed to determine the prognostic impact and cost-effectiveness ratio of preventive pharmacological interventions in these patients.

## 5. Strengths and Limitations

Our study is characterized by having an unselected and prospective sample of the general population with a CCTA study from a northwestern region of Spain, followed-up for a median of five years. On the other hand, the number of events is relatively small in accordance with that observed in several primary prevention studies in Spain [6]. While the small sample size could affect the obtained results, the comprehensiveness of the CCTA studies and the statistical analyses performed as well as the coherence with previous studies suggest that the results are reliable. The need to administer iodinated contrast to perform CCTA may reduce the applicability of the SIS score (and others) in primary prevention. However, radiation dose of a CCTA with a state-of-the-art CT scan is no longer a limiting factor given the similar radiation between a CAC and CCTA studies (between 1–mSv).

Finally, the follow-up of our cohort is shorter than the ten-year time interval projected by the REGICOR CAD risk function, so there could be underestimation of CAD events in our cohort. To take into account this limitation, we adjusted the analyses to the available follow-up.

## 6. Conclusions

CAC and SIS were associated to five-year CAD incidence, independently of classical CV risk factors in an asymptomatic population of North-Eastern Spain. CAD risk discrimination and reclassification were significantly improved by both indexes, but SIS overrode the effect of CAC.

## Figures and Tables

**Figure 1 jcm-11-00533-f001:**
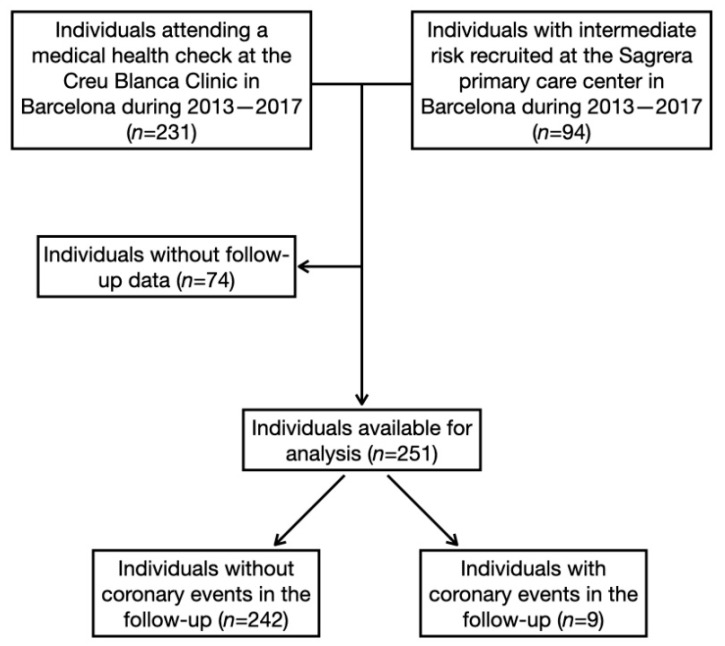
Flowchart of participants included in the study.

**Table 1 jcm-11-00533-t001:** Baseline characteristics of the cohort.

	All *n* = 251	No CAD Event in the Follow-Up *n* = 242	CAD Event in the Follow-Up *n* = 9	Standardized Differences	*p*-Value
Age (years)	58.4 (11.2)	58.3 (11.3)	60.8 (8.83)	0.245	0.525
Gender (% female)	64 (25.5%)	64 (25.5%)	0.0 (0.0%)	0.848	0.080
Total cholesterol (mgr/dL)	201 (39.7)	202 (39.4)	178 (45.0)	0.567	0.082
HDL cholesterol (mgr/dL)	53.4 (18.7)	53.8 (18.9)	42.7 (6.98)	0.782	0.045
Systolic blood pressure (mm Hg)	134 (17.7)	133 (17.4)	149 (18.4)	0.887	0.007
Dyastolic blood pressure (mm Hg)	83.1 (10.8)	82.9 (10.6)	88.4 (13.8)	0.454	0.117
Diabetes (%)	57 (23.0%)	51 (21.3%)	6 (66.7%)	1.026	0.002
Smoking (%)	72 (28.7%)	69 (28.5%)	3 (33.3%)	0.104	0.790
Framingham-REGICOR risk	12.8 (7.68)	12.5 (7.5)	20.0 (8.96)	0.905	0.006
CAC (Arbitrary units)	220 (605)	176 (427)	1232 (1980)	0.737	<0.001
Percent CAC = 0	94 (43.5%)	94 (45.4%)	0.0 (0.0%)	1.290	0.009
Segment Involvement Score (number of segments with coronary artery plaques)	2.79 (3.17)	2.59 (3.01)	7.89 (2.93)	1.782	<0.001
Percent SIS < 5	170 (73.0%)	169 (75.4%)	1 (11.1%)	1.707	<0.001

Data are shown as mean ± standard deviation except for gender, diabetes, smoking, individuals with Agatston score ≤ 0, and individuals with Segment involvement score between 0 and 4, for which the number of individuals and the percentage is presented. CAD: Coronary artery disease; HDL: high-density lipoprotein cholesterol; REGICOR: Registre Gironí del Cor/Girona Heart Registry.

**Table 2 jcm-11-00533-t002:** Discrimination improvement of the REGICOR CAD risk function including the CAC and the Segment Involvement Scores.

	Sommer’s D (95% CI)	*p*-Value *
REGICOR CAD risk function alone	0.74 (0.61, 0.87)	-
with coronary artery calcium	0.79 (0.65, 0.92)	0.074
with segment involment score	0.89 (0.83, 0.96)	0.003
with coronary artery calcium and segment involment scores	0.89 (0.83, 0.96)	0.004

* Compared to the Framingham-REGICOR CAD risk function Sommer’s D.

**Table 3 jcm-11-00533-t003:** Reclassification improvement of the REGICOR CAD risk function including the CAC and the Segment Involvement Scores.

	Continuous Net Reclassification Index (NRI)		Categorical Net Reclassification Index (NRI)	
	NRI Difference (%) * (95% CI)	*p*-Value	NRI Difference (%) * (95% CI)	*p*-Value
REGICOR CAD risk function with CAC				
Total	0.69 (0.07, 1.44)	0.034	0.06 (−0.33, 0.45)	0.773
CAD cases	−0.11 (−0.71, 0.60)	0.812	−0.11 (−0.50, 0.28)	0.577
Non-cases	0.80 (0.72, 0.87)	<0.001	0.17 (0.11, 0.22)	<0.001
Intermediate risk group	-	-	0.24 (0.05, 0.42)	0.013
REGICOR CAD risk function with Segment Involvement Score				
Total	1.14 (0.48, 1.64)	0.002	0.52 (0.17, 0.87)	0.003
CAD cases	0.56 (−0.13 1.00)	0.080	0.33 (−0.01, 0.67)	0.050
Non-cases	0.59 (0.47, 0.69)	<0.001	0.19 (0.11, 0.27)	<0.001
Intermediate risk group	-	-	0.27 (0.08, 0.45)	0.006
REGICOR CAD risk function with CAC and Segment Involvement Scores				
Total	1.17 (0.52, 1.66)	0.006	0.42 (0.12, 0.71)	0.005
CAD cases	0.56 (−0.10, 1.00)	0.090	0.22 (−0.06, 0.50)	0.122
Non-cases	0.62 (0.51, 0.72)	<0.001	0.20 (0.11, 0.28)	<0.001
Intermediate risk group	-	-	0.30 (0.12, 0.47)	0.001

* Compared to the REGICOR CAD risk function.

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
