# Peer review of "Usefulness of Cardiac Computed Tomography in Coronary Risk Prediction: A Five-Year Follow-Up of the SPICA Study (Secure Prevention with Imaging of the Coronary Arteries)"

_jcm, 2022, doi:10.3390/jcm11030533_

Round 1
Reviewer 1 Report
The authors examine the prognostic impact of CCTA derived scores, (SIS and CAC) on patients with no symptoms of CVD. The value of adding CAC, and SIS to the Framingham-REGICOR risk score is assessed in the spanish population. The modified score was more specific than traditional risk factors to improve CVD predicction and reclassification. They also reported SIS score to be a slightly better marker compared to the SIS score.
Overall the study is important, I have only 2 major comments
- CAC score does not require the full CCTA protocol whereas SIS does, thereby it becomes a poor tool for primary prevention assesment. Please highlight this point in the limitation section.
- The study is extremely underpowered, please consider toning down the discussion section, in line 247-268, given the findings are probably hypothesis generating.
Reviewer 2 Report
The authors conducted a prospective cohort study to investigate the utility of CAC and SIS in coronary risk prediction in asymptomatic individuals. The manuscript is well written. However, I have the following suggestions to help improve the clarity.
- The authors should present the cox proportional hazard analysis and adjusted HRs for CAC/SIS categories in the manuscript.
- Figure 1. The flowchart of participants is missing from the manuscript.
